# Quantifying the kinetics of hematocrit and platelet count during febrile phase to develop a scoring system for predicting dengue shock syndrome in adults: A matched case - control study from a Hospital in Viet Nam

**Mai Vu Thi Thanh**[1,2], **Hanh Bui Thi Bich**[1,2], **Vinh Ha**[1,2], **Nghia Ho Dang Trung**[1,2]*

**1** Pham Ngoc Thach University of Medicine, Ho Chi Minh City, Viet Nam, **2** Hospital for Tropical Diseases, Ho Chi Minh City, Viet Nam

* nghiahdt@pnt.edu.vn

## Abstract

### Introduction

Early prediction of dengue shock syndrome (DSS) is crucial for effective patient triage and management. However, the lack of consensus regarding the precise definition of the laboratory warning sign (WS) -"an increase in hematocrit concurrent with a rapid decrease in platelet count"- has made it difficult to utilize this WS for predicting DSS.

### Methods

A matched case - control study was conducted among adult dengue patients (aged ≥16 years) hospitalized within the first four days of illness between November 2022 and August 2023, with each DSS case was matched with three non-DSS ones.

### Results

There were 448 patients (112 DSS and 336 non-DSS) included in this study. An increase in hematocrit concurrent with a rapid decrease in platelet count was observed 1–2 days prior to the development of DSS. The cut-off value of an increase in hematocrit by ≥5% concurrent with a decrease in platelet count by ≥50% as compared with those of the previous day was found to be a predictor of DSS, with a sensitivity of 60.71% and a specificity of 83.04%. A DSS scoring system developed using these two cut-off values, along with the number of clinical WSs, can be used to predict the risk of DSS in adult patients. It achieved an area under the receiver operating characteristic curve (AUC) of 0.93 (95% CI: 0.90–0.96), sensitivity of 86.6%, and specificity of 87.8%. The Score enables triage of patients into low-, intermediate-, and high-risk groups for appropriate monitoring and management.

**Data availability statement:** All relevant data are within the manuscript and its Supporting Information files.

**Funding:** The author(s) received no specific funding for this work.

**Competing interests:** The authors have declared that no competing interests exist.

## Conclusions

The WS "an increase in hematocrit concurrent with a rapid decrease in platelet count" can be defined as "an increase in hematocrit ≥5% concurrent with a decrease in platelet count ≥50% compared to the previous day". The DSS score, developed from traditional WSs, serves as a good predictor of DSS in adult patients.

## Author summary

Dengue is a viral disease of major public health concern, with an estimated 100–400 million infections occurring each year across more than 100 countries. Due to the high number of cases, health systems in endemic areas often become overwhelmed during epidemic seasons. Developing a system to help triage patients is therefore among the top research priorities. According to the World Health Organization's 2009 guidelines, dengue is classified into dengue with or without WSs, and severe dengue. DSS accounts for approximately 80% of severe dengue cases. Dengue with WSs includes patients with varying degrees of severity and clinical outcomes. Based on data from 112 patients with DSS and 336 matched non-shock cases, we applied a novel approach to quantify the laboratory WS "an increase in hematocrit concurrent with a rapid decrease in platelet count," and subsequently developed a scoring system to stratify patients according to their risk of progressing to DSS. We created a calculator that can be used on mobile handheld devices at the bedside. Using the DSS risk score, we propose stratifying patients into low-, intermediate-, and high-risk categories to guide appropriate monitoring and management. The DSS risk score may also assist in selecting patients for dengue treatment trials in the future.

## Introduction

Dengue is a mosquito-borne infection caused by the dengue virus (DENV). The number of dengue cases reported to the World Health Organization (WHO) has increased from 50,543 cases in 2000 to 14.6 million in 2024 with more than 12,000 dengue-related deaths [1]. Most cases have been reported in tropical and subtropical areas where the virus is endemic. Dengue cases have also been reported in non-endemic areas, such as mainland Europe, making it a global health concern [2]. Approximately 1 in 5 infected individuals are symptomatic. The majority of those with symptoms experience a self-limited febrile illness that resolves without complications. A smaller proportion of patients develop complications leading to life-threatening severe dengue.

The WHO Dengue guidelines for diagnosis, treatment, prevention and control, launched in 2009 classifies dengue into three categories: dengue without WSs, dengue with WSs, and severe dengue [3]. Severe plasma leakage leading to DSS is the most frequent form of severe dengue, accounting for approximately 80% of all cases

[4]. WSs are symptoms and signs that occur in the later part of the febrile phase, signaling the possibility of progression to severe dengue. The guidelines list six clinical WSs (abdominal pain or tenderness, persistent vomiting, clinical fluid accumulation, lethargy/restlessness, liver enlargement >2 cm), and one laboratory WS (an increase in hematocrit concurrent with a rapid decrease in platelet count). While all the clinical WSs are obvious and easy to recognize, the only laboratory WS is defined only as a trend without specific numerical thresholds. How much hematocrit increases, and how to define a rapid decrease in platelet count were not specified. This leads to varied interpretations and practices of this WS among treating physicians. In different studies, an increase in hematocrit has been defined as >45%, >46%, >48%, >50% [5]. A rapid decrease in platelet count has been defined as <50 x 10⁹/L or <100 x 10⁹/L in the majority of studies, some studies used the cut-off values of <20 x 10⁹/L and even <150 x 10⁹/L. There is only one study that clearly defined the laboratory WS as an increase in hematocrit together with a decrease of >10 x 10⁹/L platelets within 24 hours compared with the previous measurement, or concurrent with a platelet count ≤100 x 10⁹/L [6]. The lack of unified criteria to define the laboratory WS has made it less useful in daily clinical practice as a predictor of severe dengue.

There are no effective antivirals for dengue; therefore, early recognition of complications is crucial for ensuring appropriate clinical management. The establishment of reliable systems to triage patients according to their risk of developing severe dengue is a major research priority [7]. This study was carried out (1) to determine the level of increase in hematocrit and the rate of decrease in platelet count between two consecutive days that should be considered as a WS, and (2) to develop a practical scoring system based in corporation all WSs to facilitate the stratification of patients into risk categories for progression to DSS, thereby enabling more appropriate clinical management.

## Methods

### Ethical statement

The study was conducted in accordance with Good Clinical Practice and the guidelines of the Declaration of Helsinki, and was approved by The Ethics Committee for Biomedical Research of The Hospital for Tropical Diseases, Viet Nam (IORG0007145) on 30 November 2022. All patients in the prospective period provided written informed consent. Patients aged 16–17 years and their parents/guardians were introduced the study information. Patients gave their assent, and their parents/guardians provided written informed consent. In the retrospective period, the data were extracted from medical records so that patients' consent was waived by the Hospital's Ethics Committee for Biomedical Research.

### Study design

This was a matched case - control study [8]. The study was carried out at the Hospital for Tropical Diseases, a tertiary referral hospital for infectious diseases in Ho Chi Minh City, Viet Nam, from November 2022 to August 2023.

### Inclusion criteria

Eligible patients were aged ≥16 years with dengue infection confirmed by a positive NS1 rapid test or an anti-dengue IgM ELISA, who were admitted to the hospital between day 1 and day 4 of illness and who underwent daily complete blood count (CBC) monitoring.

### Exclusion criteria

Patients who had undergone anti-shock management for dengue either at the time of admission or at a referring hospital, those with underlying conditions associated with thrombocytopenia or severe anemia, those without at least two CBC results available from consecutive days prior to the onset of shock (from two different illness days, with the second obtained at least 3 hours before shock onset), and those who received blood or blood product transfusions during the study period were excluded.

## Matching criteria

Each case of DSS was matched to three non-DSS cases based on age (±5 years), sex, CBCs on the same day of illness, and pregnancy status. When more than three potential non-shock controls were available, priority for selection was given to those most comparable in terms of time of hospital admission, the day of illness at admission, and the admitting ward. Matching was performed in both retrospective and prospective phases. In the retrospective phase, case identification and matching were done concurrently with data collection, while in the prospective phase matching was performed after data collection. In both phases, the matching process stopped once three suitable controls were assigned per case, with random selection based on predefined priority criteria to ensure consistency and minimize bias.

## Recruitment procedure

The study included two periods: a prospective period from December 2022 to August 2023, and a retrospective period from December 2021 to November 2022, during which data were retrieved from medical records.

All patients were treated in accordance with the national 2019 guidelines [9] and WHO 2009 guidelines [3]. Vital signs and WSs were assessed every six hours or more frequently as warranted, and CBCs were obtained daily (at admission and each morning thereafter). For patients with multiple CBC results in a single day, the earliest value was chosen. Other biochemical and imaging tests were performed at the request of treating physicians, as indicated.

## Definition of variables

- Day of illness was defined as the number of days from fever onset, with day 1 defined as the first day of fever.

- Adult was defined as patients aged ≥16 years old were treated in the adult ward according to national policy, therefore, in this study they were considered as adult.

- DSS was defined as plasma leakage leading to circulatory failure, manifested by restlessness, irritability or lethargy, cold extremities, a rapid and thready pulse, pulse pressure ≤ 20 mmHg or hypotension including undetectable pulse and blood pressure [9].

- Hematocrit increase rate (HIR) was defined as the percentage of Hct increase between two consecutive days in the febrile phase of dengue. HIR (%) = $\left( \frac{Hct_{Dn} - Hct_{Dn-1}}{Hct_{Dn-1}} \right)$ x100%.

- Platelet decrease rate (PDR) was defined as the percentage of Plt decrease between two consecutive days in the febrile phase of dengue. PDR (%) = $\left( \frac{Plt_{Dn-1} - Plt_{Dn}}{Plt_{Dn-1}} \right)$ x100%.

- Obesity was defined using the WHO Asia-Pacific standard as body mass index (BMI) ≥25 kg/m$^2$.

## Outcome measurement

The occurrence of DSS was defined as whether patients progressed to DSS or remained non-shock until discharge.

## Statistical analysis

Data were entered and analyzed using IBM SPSS Statistics version 27 (IBM Corp., New York, USA). In this matched observational study, data from the DSS and non-DSS groups were analyzed using conditional (fixed-effects) logistic regression, with p < 0.05 considered statistically significant. Regression modeling was performed with conditional stepwise estimation, and diagnostic performance was assessed in terms of sensitivity, specificity, and likelihood ratios. To enhance statistical power, we adopted a predefined 1:3 matching ratio, which resulted in a disproportionately high prevalence of DSS (25%) in the study cohort [10]. Consequently, positive likelihood ratios and negative likelihood ratios—measures

that are not influenced by prevalence—were used instead of positive and negative predictive values, which are inherently dependent on prevalence. For score development, the final regression model was transformed into a practical scoring system by multiplying the coefficients of the adjusted variables by 2 and rounded them to get the corresponding point values. The predictive model was internally validated using cross-validation and bootstrapping, while the scoring system was assessed by the receiver operating characteristic (ROC) curve and calibration, all performed using SPSS software.

## Results

During the study period, a total of 448 dengue patients were enrolled, comprising 112 DSS cases and 336 matched non-DSS controls (Fig 1).

### Baseline clinical and paraclinical characteristics

The median age of the patients was 25 years, and the male-to-female ratio was 1.3:1.

Most patients (80%) were admitted on illness days 3 and day 4. Common symptoms included fever (100%), myalgia (72%), persistent vomiting (58%), and abdominal pain or tenderness (50%). Except for fever, these symptoms were significantly more frequent in DSS patients than in non-DSS patients. There was one pregnant patient in the DSS group, who was matched with three pregnant patients in the non-DSS shock group. Obesity was significantly more common in DSS patients in univariate analysis (p < 0.05), but this association was not confirmed in multivariate analysis (p = 0.505). Among the 112 DSS patients, shock occurred most often on day 5 (55%), followed by day 6 (28%), day 4 (13%), and day 7 (4%). Clinical or radiological signs of fluid accumulation were also more common in the DSS group (42% vs. 15%, p = 0.001). Liver transaminase levels were higher in DSS patients than in non-DSS patients, with AST levels exceeding ALT levels (Table 1).

### Kinetics of Hematocrit and Platelet in DSS Compared with Non-DSS

In non-DSS patients, hematocrit rose slightly from day 2, peaked on day 5, and then declined, whereas in DSS cases, it increased more rapidly 1–2 days before shock onset. Platelet counts declined from day 2 in both groups, with a steeper drop in DSS cases on days 3–5 (S1 Fig). Compared with non-DSS patients, DSS cases showed significantly higher HIR and PDR from day 3 to day 5, but not from day 2–3 (Fig 2, Fig 3; S1 Table)

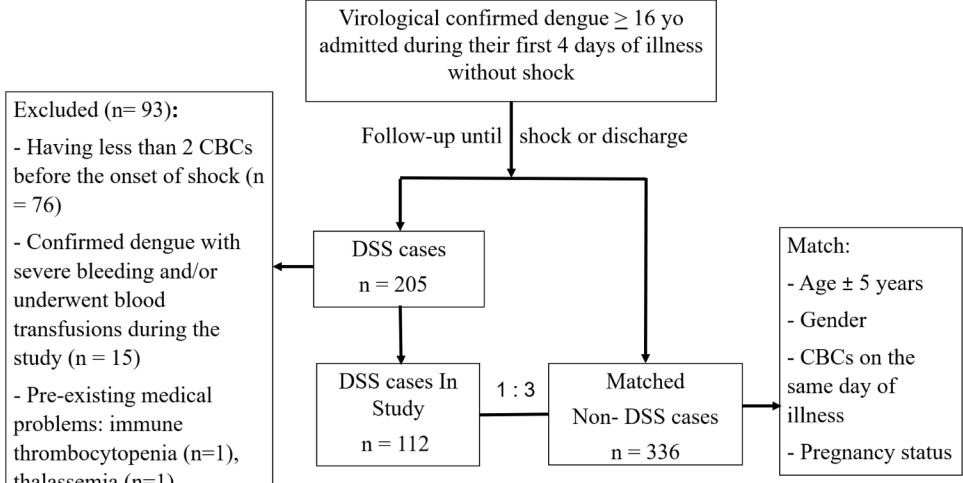

**Fig 1. Schema of the recruitment of patients into the study.**

**Table 1. Characteristics of the study population.**

| | DSS (N = 112) | Non-DSS (N = 336) | P-value (*) |
|---|---|---|---|
| **Clinical characteristics** | | | |
| **Age (years)** | 25 (21 – 31) | 25 (21 – 33) | |
| **Gender male** | 64 (57) | 192 (57) | |
| **Pregnancy** | 1 | 3 | |
| **Previous dengue** | 9 (8) | 17 (5.1) | 0.26 |
| **Underlying diseases** | 20 (17) | 47 (14) | 0.27 |
| **Admission day** | | | 0.236 |
| Day 1 | 2 (1.8) | 11 (3.3) | |
| Day 2 | 14 (12.5) | 66 (19.6) | |
| Day 3 | 49 (43.8) | 109 (32.4) | |
| Day 4 | 47 (42) | 150 (44.6) | |
| **Obesity** | 42 (38) | 89 (27) | 0.03 |
| **Fever** <br> **Duration of fever (days), mean ± SD** | 112 (100) <br> 4.29 ± 1.14 | 336 (100) <br> 4.56 ± 1.21 | |
| **Myalgia** | 92 (82) | 231 (69) | 0.05 |
| **Persistent vomiting** | 79 (71) | 179 (53) | 0.001 |
| **Abdominal pain** | 77 (69) | 144 (43) | 0.001 |
| **Hemorrhage (**)** | 91 (81) | 152 (45) | 0.001 |
| Petechiae | 71 (63) | 105 (31) | 0.001 |
| Mucosal bleed | 50 (44) | 74 (22) | 0.001 |
| **Fluid accumulation** | 47 (42) | 49 (15) | 0.001 |
| **Number of WS:** | | | |
| 0 | 14 (13) | 200 (60) | |
| 1 | 13 (12) | 87 (26) | |
| ≥ 2 | 85 (76) | 49 (15) | 0.00001 |
| **Shock onset day** | | | |
| DSS on day 4 | 15 (13.4) | | |
| DSS on day 5 | 62 (55.4) | | |
| DSS on day 6 | 31 (27.7) | | |
| DSS on day 7 | 4 (3.6) | | |
| **Paraclinical characteristics** | | | |
| AST (**) (U/L) | 154 (109 – 173) | 144 (85 – 196) | 0.01 |
| ALT (**) (U/L) | 107 (57 – 183) | 83 (57 – 156) | 0.001 |
| **Ultrasound of the abdomen (n):** | 83 | 249 | |
| Hepatomegaly | 28 (34) | 39 (16) | 0.001 |
| Thickened gallbladder wall | 49 (59) | 81 (33) | 0.001 |
| Ascites | 43 (52) | 62 (25) | 0.001 |
| Pleural effusion | 28 (34) | 43 (17) | 0.001 |

Data are n (%) or median (IQR: interquartile range) or mean ± SD (Standard Deviation) unless specified. WS: warning sign, DSS: dengue shock syndrome. (*) Conditional logistic regression. (**) AST: Aspartate aminotransferase, ALT: Alanine aminotransferase.

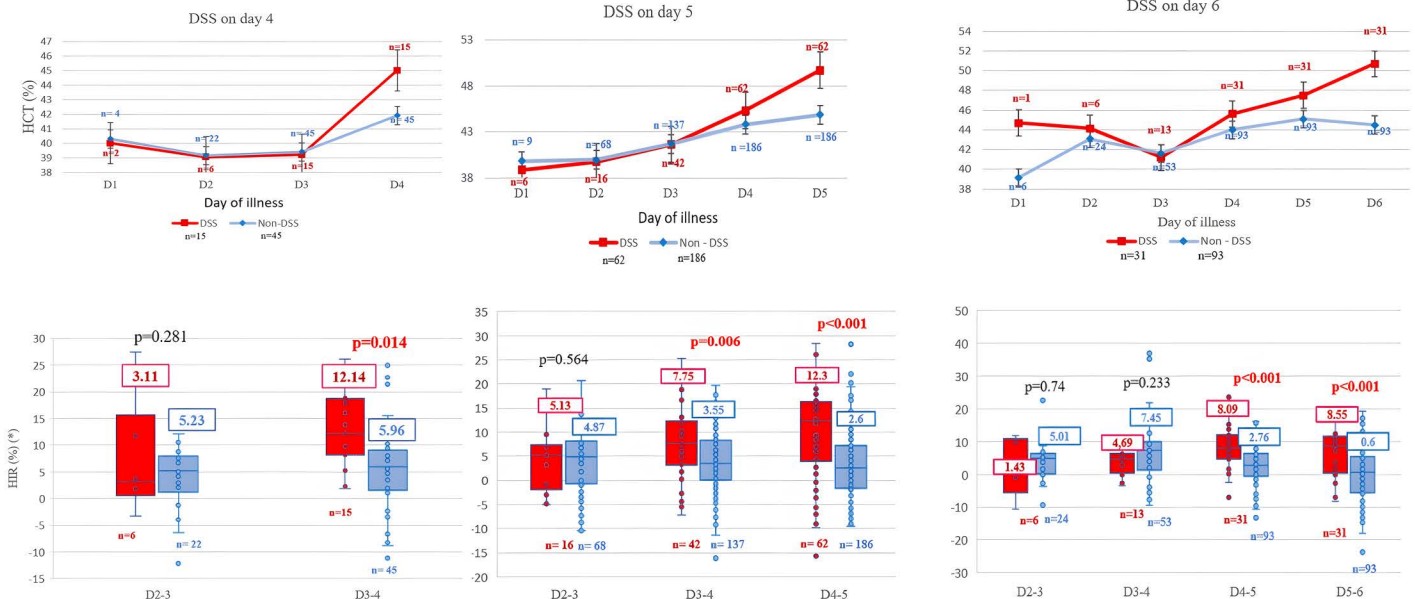

**Fig 2. Kinetics of hematocrit (above) and HIR (below) in DSS and non-DSS cases during consecutive days of illness before shock onset.** (HCT: hematocrit, HIR: hematocrit increase rate, DSS: dengue shock syndrome).

## Diagnostic value of HIR and PDR in predicting DSS

Based on the Youden index, we have chosen cut-off values of HIR ≥ 5% and PDR ≥ 50%. At these thresholds, the AUCs were 0.79 (95% CI: 0.74–0.84) for HIR and 0.81 (95% CI: 0.78–0.86) for PDR. Sensitivity, specificity, LR +, and LR– were 78.6%, 63.4%, 2.15, and 0.34 for HIR, and 78.6%, 72.9%, 2.9, and 0.29 for PDR, respectively (S2 Fig). The combination of HIR ≥ 5% and PDR ≥ 50% yielded an AUC of 0.81 (95% CI: 0.77–0.85), with sensitivity of 69.7%, specificity of 83.0%, LR+ of 3.6, and LR– of 0.47 (Fig 4).

In the multivariate analysis, only HIR, PDR, and the number of clinical WSs remained significantly associated with DSS (S2 Table). Their combined ROC analysis showed excellent performance, with an AUC of 0.93 (95% CI: 0.90–0.96), sensitivity of 86.6%, specificity of 87.8%, LR+ of 7.1, and LR– of 0.15 (Fig 4). The logistic regression model incorporating HIR, PDR, and the number of WSs was internally validated using 5-fold cross-validation, achieving a mean accuracy of 87.4% (SD = 4.5), sensitivity of 69.7% (SD = 15.1), specificity of 93.0% (SD = 2.9), and AUC of 0.93 (SD = 0.01). Bootstrap resampling (5,000 iterations) confirmed the robustness of the predictors, with HIR (B = 0.140, p < 0.001), PDR (B = 0.035, p = 0.003), and WSs (B = 1.094, p < 0.001) remaining significant.

## Prediction score for Dengue shock syndrome

Multivariate regression identified three independent predictors of DSS (p < 0.001): HIR ≥ 5% (coefficient = 1.59), PDR ≥ 50% (coefficient = 1.91), and ≥ 2 clinical warning signs (coefficient = 3.06) (S3 Table). For ease of clinical application, regression coefficients were rounded and doubled to create a simplified DSS scoring system, enhancing memorability and facilitating routine use [11]. These were converted into a point-based score: 3 points for HIR ≥ 5%, 4 points for PDR ≥ 50%, and 6 points for ≥ 2 WSs. Table 2 shows two groups at the extremes: a group with LR+ > 10 (DSS score 9, 10, 13) which strongly indicates a high probability of progression to shock, and another with LR+ at around 2 and LR- < 0.1 (DSS score 3, 4), suggesting a low risk of developing shock. Therefore, the risk of DSS based on our scoring system can be stratified into three categories: low risk (≤4 points), intermediate risk (6–7 points), and high risk (≥9 points). (Table 2).

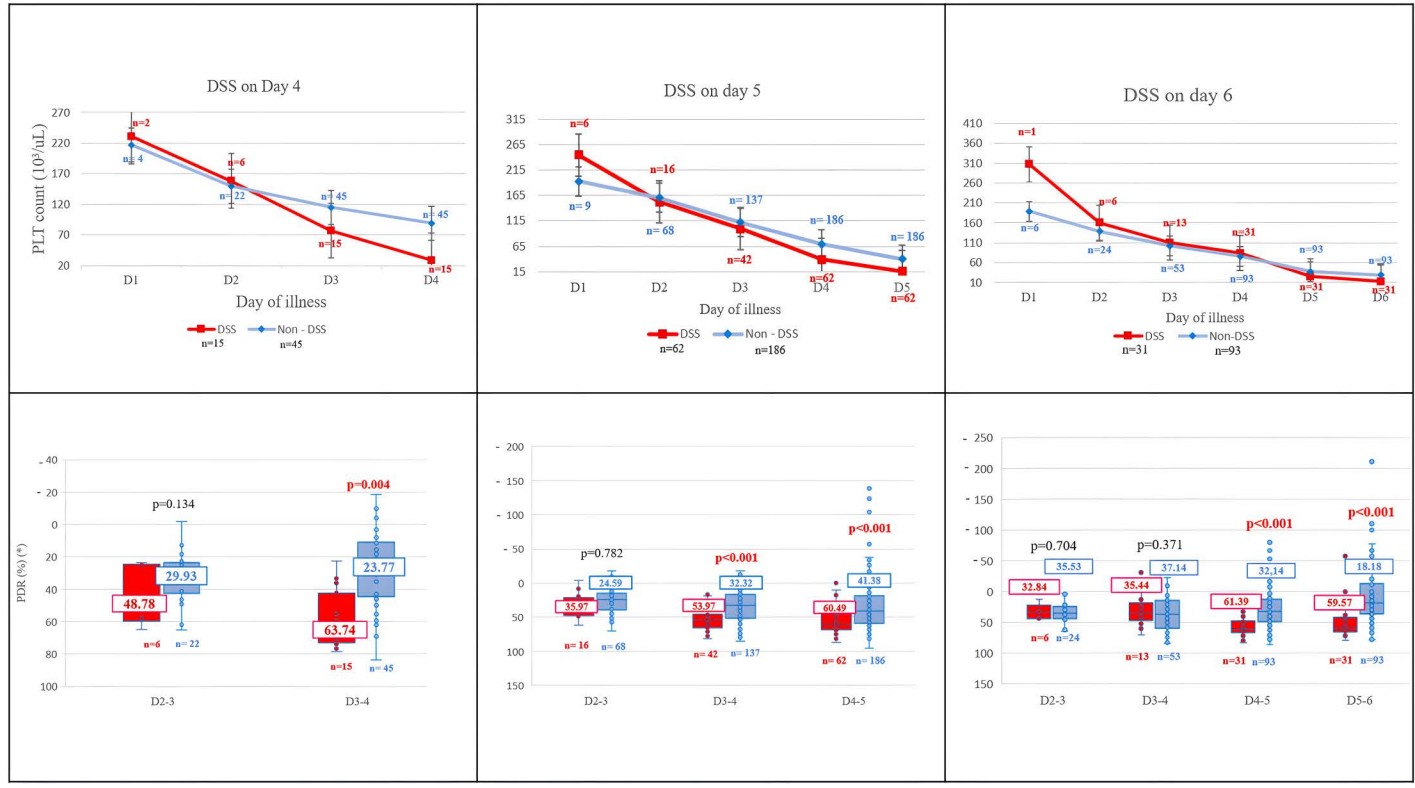

**Fig 3. Kinetics of platelet count (above) and PDR (below) in DSS and non-DSS cases during consecutive days of illness before shock onset shock onset.** (PLT: platelet, PDR: platelet decrease rate, DSS: dengue shock syndrome, 10³/μL = 10⁹/L).

The DSS score showed excellent discrimination with an AUC of 0.91 (S3 Fig). Calibration analysis demonstrated close agreement between predicted probabilities and observed outcomes, with data points clustering near the diagonal line (S4 Fig). These findings indicate that the score is both highly discriminative and well-calibrated for predicting DSS.

## Discussion

The median age of our patients was 25 years, which was younger than that reported in studies from Singapore or Taiwan [12,13]. The median BMI was 24 kg/m² in the DSS group and 22 kg/m² in the non-DSS group (p = 0.07). Fewer than 20% of patients had comorbidities (diabetes, hypertension, or liver disease), and prior dengue infection was rarely reported.

In this matched case-control study, most DSS events (95%) occurred between illness days 4 and 6, peaking on day 5. This highlights the need for close monitoring of clinical and laboratory parameters from illness days 2–3 onward to enable early recognition of signs predicting dengue shock. Hematocrit, reflecting plasma leakage, rose significantly faster in DSS patients compared with non-DSS patients beginning 1–2 days before shock, as captured by the HIR. Thrombocytopenia, a well-recognized feature of dengue, likely results from both reduced bone marrow production and increased peripheral destruction of platelets [14]. In our cohort, platelet counts began to decline on day 3 and reached their lowest levels on day 6 in both DSS and non-DSS patients, consistent with previous reports and supporting their proposed role in plasma leakage and inflammation.

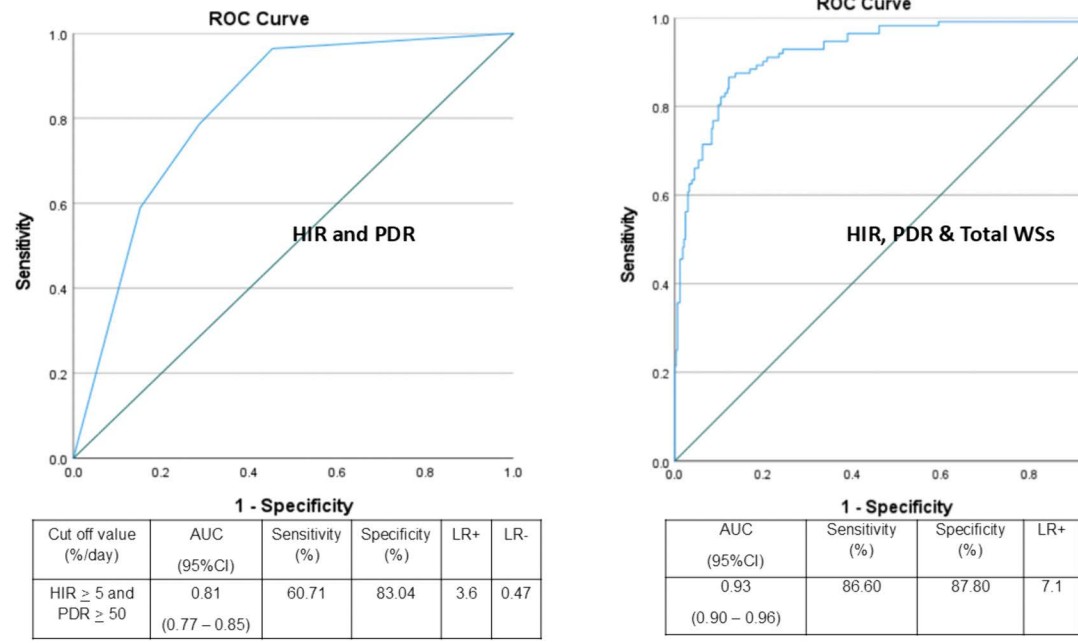

**Fig 4. Receiver operating characteristic curve showing the performance of HIR plus PDR (left panel), and of the combination of HIR, PDR and number of clinical WSs (right panel).** HIR: hematocrit increase rate, PDR: platelet decrease rate, ROC: receiver operating characteristic curve, AUC: area under the ROC curve, LR: likelihood ratio.

**Table 2. Proposed stratification of DSS risk based on DSS score.**

| DSS Score | Sensitivity (%) | Specificity (%) | LR+ | LR- | Risk level |
|---|---|---|---|---|---|
| 3 | 98.21 | 45.53 | 1.81 | 0.04 | Low |
| 4 | 96.64 | 61.01 | 2.43 | 0.09 | |
| 6 | 89.29 | 72.62 | 3.26 | 0.15 | Intermediate |
| 7 | 87.50 | 81.55 | 4.74 | 0.15 | |
| 9 | 74.11 | 94.35 | 13.11 | 0.27 | High |
| 10 | 59.82 | 95.83 | 14.37 | 0.42 | |
| 13 | 45.54 | 97.62 | 19.13 | 0.56 | |

DSS: dengue shock syndrome, LR: likelihood ratio.

The WHO 1997 guidelines defined dengue hemorrhagic fever by a hematocrit increase of ≥20% and a platelet count <100 x 10⁹/L, while the WHO 2009 guidelines only considered rising hematocrit concurrent with rapidly falling platelets as a laboratory WS, without specifying numerical thresholds. There was one study following the dynamics of hematocrit and platelet count for predicting DSS in children aged 5–15 years [15]. Lam et al. analyzed data from 2,301 patients, of whom 143 developed shock (6%). In a smaller cohort of 908 patients enrolled on day 3 of illness, the researchers used a logistic regression model and graphical analysis to assess the predictive value of daily hematocrit and platelet count for DSS. The study showed that serial platelet counts strongly predicted DSS, whereas daily hematocrit values contributed little [15]. In our adult population (n = 448), HIR and PDR differed significantly between DSS and non-DSS patients 1–2 days before the onset of shock. Our findings are consistent with Lam et al.'s study regarding platelet kinetics but differ in hematocrit performance. This discrepancy may reflect both physiological differences in hematocrit dynamics between adults and

children and methodological factors such as Lam's inclusion of severe bleeding cases, which may lower hematocrit values. Differences in study design may also contribute: in our study, each case was matched with three controls, whereas in Lam's study, the DSS group was compared with the rest of the study population. The imbalance between cases and controls may obscure the true positive effect of hematocrit on DSS, which is smaller in magnitude than that of platelet count. Moreover, in graphical analysis, the authors primarily compared absolute values of serial hematocrits in DSS cases versus those in non-DSS cases, while we used dynamic indices (HIR) rather than absolute values. The combination of HIR ≥ 5% and PDR ≥ 50% predicted DSS with AUC of 0.81 (95%CI: 0.77–0.85), sensitivity of 60.7%, specificity of 83.0%, and positive likelihood ratio of 3.6, indicating that patients meeting these cut-offs were 3.6 times more likely to develop shock than those who did not. These findings support HIR ≥ 5% and PDR ≥ 50% as practical laboratory WS thresholds for early identification of dengue patients at risk of DSS.

Many scoring systems have been proposed to predict severe dengue, using either admission clinical and laboratory data or sophisticated biomarkers. Pongban et al. developed a scoring system based on six parameters: age, hepatomegaly, hematocrit, systolic blood pressure, white blood cell count, and platelet count. In their study, 39 out of 90 DSS cases had a score >11.5 (with scores ranging from 0 to 18) [16]. Based on data from 302 dengue-infected patients in Thailand, Srisuphanunt et al. constructed a scoring system for early prediction of severe dengue using six laboratory tests, with a cut-off score of 14 to predict DSS (score range: 0 to 38.6) [17]. Researchers in Viet Nam used various machine learning models to analyse data from 230 pediatric dengue patients to develop a nomogram for predicting DSS risk [18]. The nomogram used five parameters: albumin, activated partial thromboplastin time, fibrinogen, aspartate aminotransferase, and platelet count, with scores ranging from 0 to 350 points. The model performed well in both training and validation sets, with AUC of 0.985 (95% CI: 0.965-1.000) and accuracy of 0.988 (95% CI: 0.957-0.998) in the training set, and AUC of 0.945 (95% CI: 0.886-1.000) with accuracy of 0.951 (95% CI: 0.865-0.989) in the validation set [18]. The above-mentioned studies required many non-basic laboratory tests, which may not be available at the bedside in endemic areas. A study on adults with dengue in Taiwan used four parameters to develop a severity risk score for dengue patients with ≤4 days of illness, and two parameters for those with >4 days of illness. At the cut-off value of 1 point, the sensitivity and specificity were estimated at 70.3% and 90.6%, respectively [19]. The patients in this study had a median age of 51 years in the non-severe group and 66 years in the severe dengue group, which is much higher than that of our patients. Recently, Madewell and colleagues used various machine learning models to analyze data from dengue cases in Puerto Rico for predicting severe dengue [20]. Among the 1,708 laboratory-confirmed cases, 415 were classified as severe dengue. Due to an imbalance in the dataset, with non-severe cases being more prevalent, the researchers applied up-sampling to balance the class distribution. The ensemble model using 40 variables achieved the highest overall AUC of 0.977 (with sensitivity of 95.6% and specificity of 93.3%). The performance of each WS in predicting severe dengue was also studied using a logistic regression model. The presence of any WSs yielded the highest sensitivity (92.8%), but low specificity (29.2%), with an AUC of 0.611. Combining ≥3 WSs resulted in an AUC of 0.713, with sensitivity of 87.2% and specificity of 65.1% (LR+ of 2,5, LR- of 0.37). Despite this high predictive accuracy of machine learning models, the authors acknowledged that their implementation in clinical practice may require computational resources that may not be available in dengue-endemic settings. Therefore, to enhance clinical utility, logistic regression models could complement machine learning approaches by enabling clinicians to apply these findings more feasibly in practice [20]. Our study in adults employed daily bedside clinical WSs combined with two laboratory WSs derived from routine CBC results. Multivariate analysis identified three independent predictors of DSS: ≥ 2 clinical WSs, HIR ≥ 5%, and PDR ≥ 50%. A reduced logistic regression model incorporating these factors showed excellent predictive performance (AUC = 0.93, 95% CI: 0.90–0.96). There are two main differences between the study by Madewel et al. and our own that may explain why our approach achieved a higher AUC. First, our study focused only on DSS, while Madewel et al. included severe bleeding, a condition in which hematocrit usually decreases rather than increases. This may have interfered with the positive effect of hematocrit

increase in predicting DSS. Second, our study included platelet count reduction and fluid accumulation in the WS list, while Madewel *et al.* excluded them.

Based on the sensitivity, specificity, positive and negative likelihood ratios of the DSS score which depend on the probability of progressing to shock, we empirically derived a practical framework to triage dengue patients into three DSS risk groups: low, intermediate and high risk (Table 2). From a clinician's perspective, this framework provides actionable guidance for monitoring frequency tailored to the estimated risk, and the DSS score can be applied as early as day 2 or day 3 of illness to ensure timely triage and optimize clinical decision-making. Very low-risk patients (score 0), who have a minimal probability of developing shock, may be managed as outpatients and advised to return the following day or sooner if new WSs occur. Low-risk patients could be monitored every 6–12 hours, while intermediate-risk patients may require closer observation every 3–6 hours. High-risk patients should have vital signs checked every 30–60 minutes to detect early progression to shock. The score should be recalculated whenever new WSs or laboratory results become available, ensuring that risk stratification remains dynamic and responsive to clinical changes. The DSS score demonstrated excellent discrimination (AUC = 0.91) and good calibration, with predicted risks closely matching observed outcomes. This framework is intended to support clinical decision-making and optimize resource allocation, particularly in settings with high patient loads. While our recommendations are empirically supported by the dataset, they should be validated in larger, prospective studies before being adopted into routine practice.

Our study has several limitations. We lacked data on dengue virus serotypes, viral load, and IgG/IgM status, which may influence the risk of shock. However, by using a matching method and recruiting patients during a single epidemic season, the differences between two groups may have been minimized. Although the sample size was not pre-calculated, our cohort of 112 DSS and 336 non-DSS patients could provide sufficient power for analysis. Requiring at least two consecutive CBCs increased workload and treatment costs, which may limit the applicability of our DSS score in some places. The Pan America Health Organization (PAHO) 2022 guidelines [21] and the WHO 2025 guidelines for clinical management of arboviral diseases [22] do not consider thrombocytopenia as a WS, since it is not a consequence of extravasation, and therefore is not considered a useful guide for medical professionals in the management of parenteral fluids in dengue. Our data support the use of changes in hematocrit and platelet count over time as laboratory WSs by calculating HIR and PDR to develop a DSS prediction score, not to guide fluid administration. Although internal validation demonstrated the good predictive ability of the DSS score, external validation in independent cohorts is lacking. The three-parameter model may risk overfitting, and the single-center, retrospective design limits generalizability. Potential inter-observer variability in WSs, effects of laboratory fluctuations on HIR/PDR, and selection bias from CBC-based matching further constrain representativeness. These limitations highlight the need for prospective validation before routine clinical use.

Our study has several strengths. First, the dynamic approach to hematocrit and platelet count by comparing each patient's hematocrit and platelet count over two consecutive days - depicts individual changes more precisely than comparisons with population baseline values. Second, the DSS score is simple and easy to use, either through mental calculation or via a small spreadsheet compatible with smart devices for rapid bedside risk assessment (available at https://doi.org/10.5281/zenodo.18513829 or https://sites.google.com/pnt.edu.vn/infectiousdiseases-dss-score, see example in S5 Fig). Thirdly, because dengue with WSs includes patients with a wide range of severity and outcomes - from mild cases that never progress to shock to critical dengue requiring ICU admission or resulting in death [23], our triage schema stratifies patients into three risk levels. This allows redistributing of clinical resources toward high-risk patients while reducing unnecessary workload for low-risk patients. This approach is more practical than the current recommendation of uniformly monitoring vital signs every 2–4 hours for all patients with WS [24].

To minimize selection bias, we followed patients from admission with non-severe dengue until DSS or recovery, excluding those with prior anti-shock treatment, transfusions, or hematological disorders. This allowed us to focus on the natural course, with shock as the predominant severe manifestation; future studies should examine predictors of other severe forms such as hemorrhage, marked liver enzyme elevations, and organ involvement.

## Conclusions

We propose that the WHO 2009 laboratory WS for dengue could be specified as "an increase in hematocrit ≥5% concurrent with a decrease in platelet count ≥50% compared with those of the previous day". The DSS Risk Score enables triage of patients into low-, intermediate-, and high-risk groups for appropriate monitoring and management, with early use from day 2 or 3 of illness allowing prompt risk assessment and timely clinical decisions. It may also guide patient selection in future clinical trials of DSS therapies. Further studies in other settings, outpatient settings, and across different age groups, including children aged <16 years and adults aged >60 years, are needed to validate the score before broader clinical implementation.

## Supporting information

**S1 Fig. Kinetics of hematocrit and platelet count in DSS vs non-DSS.**
(TIF)

**S2 Fig. Receiver operating characteristic (ROC) curve showing the performance of HIR, and PDR in predicting DSS.**
(TIF)

**S3 Fig. Receiver operating characteristic (ROC) curve for internal validation of the DSS score.**
(TIF)

**S4 Fig. Calibration curve showing agreement between predicted and observed probabilities of DSS.**
(TIF)

**S5 Fig. Example of DSS score calculator in a smart-device compatible spreadsheet.**
(TIF)

**S1 Table. Kinetics of HIR and PDR in DSS cases according to day first occurrence of shock.**
(DOCX)

**S2 Table. Factors associated with the development of DSS.**
(DOCX)

**S3 Table. Point-based score of three independent predictors of DSS identified by multivariate regression.**
(DOCX)

## Acknowledgments

We thank all patients for allowing us to use their data for the study. We appreciate all staffs of adult dengue wards for their hardworking caring for the patients. We thank head doctors of adult wards in the hospital for helping in data collecting. We are particularly in debt to Prof. Tran Tinh Hien, Prof. Dong Thi Hoai Tam, and Dr. Phan Tu Qui for their fruitful comments.

**Conference statement:** A preliminary version of the study was presented at The Viet Nam National Scientific Conference on Infectious Diseases and HIV/AIDS, Ha Noi, 31 October - 2 November, 2024. DOI: https://doi.org/10.59873/vjid.v3i47

## Author contributions

**Conceptualization:** Mai Vu Thi Thanh, Hanh Bui Thi Bich, Vinh Ha, Nghia Ho Dang Trung.

**Data curation:** Mai Vu Thi Thanh.

**Formal analysis:** Mai Vu Thi Thanh, Nghia Ho Dang Trung.

**Funding acquisition:** Nghia Ho Dang Trung.

**Investigation:** Mai Vu Thi Thanh, Hanh Bui Thi Bich, Vinh Ha, Nghia Ho Dang Trung.

**Methodology:** Mai Vu Thi Thanh, Hanh Bui Thi Bich, Vinh Ha, Nghia Ho Dang Trung.

**Project administration:** Vinh Ha, Nghia Ho Dang Trung.

**Resources:** Nghia Ho Dang Trung.

**Software:** Mai Vu Thi Thanh, Vinh Ha.

**Supervision:** Vinh Ha, Nghia Ho Dang Trung.

**Validation:** Mai Vu Thi Thanh.

**Visualization:** Mai Vu Thi Thanh, Vinh Ha.

**Writing – original draft:** Mai Vu Thi Thanh, Vinh Ha.

**Writing – review & editing:** Mai Vu Thi Thanh, Hanh Bui Thi Bich, Vinh Ha, Nghia Ho Dang Trung.

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
