## [Decision Letter · Decision Letter 0]

24 Dec 2025

PNTD-D-25-01958

Quantifying the Kinetics of Hematocrit and Platelet Count During Febrile Phase to Develop a Scoring System for Predicting Dengue Shock Syndrome in Adults: A Matched-Case Observational Study from a Hospital in Viet Nam

Dear Dr. Ho Dang Trung,

Thank you for submitting your manuscript to PLOS Neglected Tropical Diseases. After careful consideration, we feel that it has merit but does not fully meet PLOS Neglected Tropical Diseases's publication criteria as it currently stands. Therefore, we invite you to submit a revised version of the manuscript that addresses the points raised during the review process.

Please submit your revised manuscript within by Feb 22 2026 11:59PM. If you will need more time than this to complete your revisions, please reply to this message or contact the journal office at plosntds@plos.org. Please include the following items when submitting your revised manuscript:

We look forward to receiving your revised manuscript.

Kind regards,

Mohammad Jokar, DVM

Guest Editor

Sujatha Sunil

Section Editor

Shaden Kamhawi

co-Editor-in-Chief

Paul Brindley

co-Editor-in-Chief

**Additional Editor Comments:**

• Matched case-control labeled as “matched-case observational study” is unclear—standard terminology is matched case-control. The tertiary referral hospital setting may introduce selection bias toward severe cases, so please check.

• Exclusions for prior anti-shock treatment, underlying thrombocytopenia/anemia, transfusions, and insufficient CBCs are logical to isolate natural progression; however, they exclude many severe cases, potentially underestimating risk factors.

• Matching on age (±5 years), sex, same-day CBCs, and pregnancy is strong for confounding control. Additional prioritization by admission time, ward, and day is good. The 1:3 ratio is efficient, but multiple candidates introduce selection subjectivity.

• Adherence to 2019 Vietnam and 2009 WHO guidelines is consistent, though the 2009 WHO guideline is outdated. If possible, please update it.

• Please add details about the ethical statement process for participants aged 16–17 years.

• Using Day 1 as fever onset is standard; defining ≥16 years as adult per policy is pragmatic. The DSS definition aligns with guidelines (narrow pulse pressure, cold extremities, etc.), but clinical diagnosis without echo confirmation of leakage may miss compensated shock.

• Conditional logistic regression is correct for the matched design; however:

• Stepwise selection risks overfitting/instability in small samples.

• Use of likelihood ratios over predictive values is wise given artificial 25% prevalence.

• Score development by doubling/rounding coefficients is common but arbitrary. Validation (internal/external) is missing.

• No sample size/power calculation reported.

• No adjustment for multiple comparisons or handling of missing data described.

**Journal Requirements:**

At this stage, the following Authors/Authors require contributions: Mai Vu Thi Thanh, Hanh Bui Thi Bich, Vinh Ha, and Nghia Ho Dang Trung. Please ensure that the full contributions of each author are acknowledged in the "Add/Edit/Remove Authors" section of our submission form.

2) Kindly revise your competing statement in the online submission form to align with the journal's style guidelines: 'The authors declare that there are no competing interests.'

**Reviewers' Comments:**

Reviewer's Responses to Questions

**Key Review Criteria Required for Acceptance?**

**Methods**

-Are the objectives of the study clearly articulated with a clear testable hypothesis stated?

-Is the study design appropriate to address the stated objectives?

-Is the population clearly described and appropriate for the hypothesis being tested?

-Is the sample size sufficient to ensure adequate power to address the hypothesis being tested?

-Were correct statistical analysis used to support conclusions?

-Are there concerns about ethical or regulatory requirements being met?

Reviewer #1: The study design is appropriate and sample size is OK

Reviewer #2: Dengue fever, a global public health challenge, infects approximately 100 to 400 million people annually. The massive number of cases often overwhelms local health systems during epidemic seasons, making early prediction of dengue shock syndrome (DSS) a critical issue. However, the lack of consensus on the definition of the laboratory warning sign "elevated hematocrit with rapid thrombocytopenia" complicates the use of all warning signs for DSS prediction. In this study, Vu Thi Thanh Ma et al developed a risk score system for predicting the risk of adult dengue shock syndrome (DSS) by quantifying the changes of hematocrit and platelet count during the fever stage of dengue fever patients, which can provide a scientific basis for clinical triage and management.

A key challenge involves designing matched case-control studies to ensure data reliability and result generalizability while minimizing confounding factors' impact on analysis.In the methodology section, it is necessary to describe in detail why the case-control ratio was chosen.

Reviewer #3: (No Response)

Reviewer #4: How do you analyze the blood examinations for a patient who had more than one result in a single day? Did you consider it as a mean of representing that day? (D2, D3, etc).

What about data you collected retrospectively? How did you record the diagnosis of DSS? Was it based on the medical record without any further analysis? For example, after analyzing the case carefully, you found that the patient actually did not experience shock.

**Results**

-Does the analysis presented match the analysis plan?

-Are the results clearly and completely presented?

-Are the figures (Tables, Images) of sufficient quality for clarity?

Reviewer #1: The results are clearly presented and there are clarity in figures etc.

Reviewer #2: It is necessary to describe the course days of the two groups ( DSS group and non-DSS group ) in results.The picture content of Fig.2 and Fig.3 is inconsistent with the description of the caption. The small title in the figure shows the parameter values of a day after the onset, and the content expressed is the continuous change of parameter values of different days.

We cannot know when the two parameters are used for differential diagnosis of DSS.The description of the results did not explain when the parameters were used to diagnose DSS after the onset of the patient, and how much time to diagnose DSS earlier than other methods.

Reviewer #3: (No Response)

Reviewer #4: According to your research, when is the scoring system applicable? Is the scoring system applicable on the second, third, or fourth day? Do you believe that sensitivity and specificity could vary? Please discuss this.

**Conclusions**

-Are the conclusions supported by the data presented?

-Are the limitations of analysis clearly described?

-Do the authors discuss how these data can be helpful to advance our understanding of the topic under study?

-Is public health relevance addressed?

Reviewer #1: Conclusions are supported by data

Reviewer #2: The median age of adult patients in this study was 25 years, younger than those in Singapore or Taiwan studies. Most DSS events (95%) occurred between days 4 to 6 of the illness, peaking on day 5. This underscores the importance of closely monitoring clinical and laboratory parameters starting from days 2-3. Hematocrit levels rose significantly faster in DSS patients than in non-DSS patients 1-2 days before shock onset. Thrombocytopenia, a recognized feature of dengue fever, may result from reduced bone marrow production and increased peripheral platelet destruction.

Reviewer #3: (No Response)

Reviewer #4: The conclusions are supported by the data.

**Editorial and Data Presentation Modifications?**

Reviewer #1: In line 152; please mention ( PltDn−1− Plt Dn), after”in the febrile illness. It means the line 152 should be: “in the febrile ( PltDn−1− Plt Dn) phase. PDR (%) =( PltDn−1− Plt Dn PltDn−1) x100%

Reviewer #2: The study has several limitations. Firstly, data on dengue virus serotypes, viral load, and IgG/IgM status were lacking, which may influence the risk of shock and the author should provide the above information.

Secondly, although the cohort of 112 DSS and 336 non-DSS patients provided sufficient power for analysis, the sample size was not pre-calculated . Thirdly, requiring at least two consecutive complete blood count (CBC) results increased workload and treatment costs, which may limit the applicability of the DSS score in some settings . Fourthly, the study was single-centered, and the scoring system has not yet been validated .

Reviewer #3: (No Response)

Reviewer #4: (No Response)

**Summary and General Comments**

Reviewer #1: I have reviewed the paper. Dengue is an important arboviral disease that continues to spread to newer areas, causing significant morbidity and even mortality. The World Health Organization developed guidelines for dengue in 2009, outlining various stages of the disease, including Dengue Fever, Dengue Hemorrhagic Fever (DHF), and Dengue Shock Syndrome (DSS). The present study recommends including specific parameters related to hematocrit and platelet count in the WHO classification of DSS in its 2009 manual.

Reviewer #2: The study has several limitations. Firstly, data on dengue virus serotypes, viral load, and IgG/IgM status were lacking, which may influence the risk of shock and the author should provide the above information.

Secondly, although the cohort of 112 DSS and 336 non-DSS patients provided sufficient power for analysis, the sample size was not pre-calculated . Thirdly, requiring at least two consecutive complete blood count (CBC) results increased workload and treatment costs, which may limit the applicability of the DSS score in some settings . Fourthly, the study was single-centered, and the scoring system has not yet been validated .

Reviewer #3: (No Response)

Reviewer #4: The finding is significant and can assist physicians in forecasting the occurrence of DSS.

PLOS authors have the option to publish the peer review history of their article (what does this mean?). If published, this will include your full peer review and any attached files.). If published, this will include your full peer review and any attached files.). If published, this will include your full peer review and any attached files.). If published, this will include your full peer review and any attached files.

...

Reviewer #1: No

Reviewer #2: No

Reviewer #3: No

Reviewer #4: No

**Figure resubmission:**
---

## [Decision Letter · Decision Letter 1]

19 Feb 2026

PNTD-D-25-01958R1Quantifying the Kinetics of Hematocrit and Platelet Count During Febrile Phase to Develop a Scoring System for Predicting Dengue Shock Syndrome in Adults: A Matched Case-Control Study from a Hospital in Viet NamPLOS Neglected Tropical Diseases Dear Dr. Ho Dang Trung,Thank you for submitting your manuscript to PLOS Neglected Tropical Diseases. After careful consideration, we feel that it has merit but does not fully meet PLOS Neglected Tropical Diseases's publication criteria as it currently stands. Therefore, we invite you to submit a revised version of the manuscript that addresses the points raised during the review process.Please submit your revised manuscript by Mar 21 2026 11:59PM. If you will need more time than this to complete your revisions, please reply to this message or contact the journal office at plosntds@plos.org. Please include the following items when submitting your revised manuscript:* A letter that responds to each point raised by the editor and reviewer(s). You should upload this letter as a separate file labeled 'Response to Reviewers'. This file does not need to include responses to any formatting updates and technical items listed in the 'Journal Requirements' section below.'. This file does not need to include responses to any formatting updates and technical items listed in the 'Journal Requirements' section below.* A marked-up copy of your manuscript that highlights changes made to the original version. You should upload this as a separate file labeled ''. This file does not need to include responses to any formatting updates and technical items listed in the 'Journal Requirements' section below.'. This file does not need to include responses to any formatting updates and technical items listed in the 'Journal Requirements' section below.* A marked-up copy of your manuscript that highlights changes made to the original version. You should upload this as a separate file labeled 'Revised Manuscript with Track Changes'.'.* An unmarked version of your revised paper without tracked changes. You should upload this as a separate file labeled ''.'.* An unmarked version of your revised paper without tracked changes. You should upload this as a separate file labeled 'Manuscript'.'.If you would like to make changes to your financial disclosure, competing interests statement, or data availability statement, please make these updates within the submission form at the time of resubmission. Guidelines for resubmitting your figure files are available below the reviewer comments at the end of this letter.We look forward to receiving your revised manuscript.Kind regards,Mohammad Jokar, DVMGuest EditorPLOS Neglected Tropical DiseasesSujatha SunilSection EditorPLOS Neglected Tropical Diseases'.'.If you would like to make changes to your financial disclosure, competing interests statement, or data availability statement, please make these updates within the submission form at the time of resubmission. Guidelines for resubmitting your figure files are available below the reviewer comments at the end of this letter.We look forward to receiving your revised manuscript.Kind regards,Mohammad Jokar, DVMGuest EditorPLOS Neglected Tropical DiseasesSujatha SunilSection EditorPLOS Neglected Tropical Diseases

Shaden Kamhawi

co-Editor-in-Chief

Paul Brindley

co-Editor-in-Chief

**Reviewers' comments:**Reviewer's Responses to QuestionsReviewer's Responses to Questions

**Key Review Criteria Required for Acceptance?**

**Methods:**

-Are the objectives of the study clearly articulated with a clear testable hypothesis stated?

-Is the study design appropriate to address the stated objectives?

-Is the population clearly described and appropriate for the hypothesis being tested?

-Is the sample size sufficient to ensure adequate power to address the hypothesis being tested?

-Were correct statistical analysis used to support conclusions?

-Are there concerns about ethical or regulatory requirements being met?

Reviewer #2: The study’s objectives are clearly defined and operationalized through specific analytical steps. Although a formal hypothesis is not explicitly stated, the research design and statistical tests implicitly test hypotheses about the predictive value of HIR, PDR, and their combination with clinical WS. This aligns with the standards of observational studies, where hypotheses are often framed as research questions rather than formal null/alternative hypotheses.

The matched case-control design is appropriate for addressing the study’s objectives, as it efficiently isolates the relationship between dynamic laboratory changes (HIR/PDR) and DSS risk while controlling for key confounders. However, the single-center, partially retrospective design and lack of external validation highlight the need for cautious interpretation and future multicenter studies to confirm generalizability

The population is clearly defined, well-suited to isolate the relationship between dynamic laboratory changes and DSS risk, and thus appropriate for the study’s objectives.

Reviewer #4: (No Response)

**Results**

-Does the analysis presented match the analysis plan?

-Are the results clearly and completely presented?

-Are the figures (Tables, Images) of sufficient quality for clarity?

Reviewer #2: The analysis strictly followed the pre-defined plan, with methods and results coherently aligned to address the study objectives. This consistency strengthens the reliability of the findings.

The results are presented in a clear, structured manner with comprehensive data, statistical details, and visual aids, ensuring transparency and reproducibility.

Reviewer #4: (No Response)

**Conclusions**

-Are the conclusions supported by the data presented?

-Are the limitations of analysis clearly described?

-Do the authors discuss how these data can be helpful to advance our understanding of the topic under study?

-Is public health relevance addressed?

Reviewer #2: The conclusions are tightly aligned with the study’s results, with robust statistical evidence and validation supporting both the laboratory WS definition and the DSS scoring system.

The limitations are thoroughly and clearly described, with specific attention to methodological constraints, data gaps, and generalizability. This transparency strengthens the credibility of the study by demonstrating awareness of potential biases and areas for future improvement.

The limitations are thoroughly and clearly described, with specific attention to methodological constraints, data gaps, and generalizability. This transparency strengthens the credibility of the study by demonstrating awareness of potential biases and areas for future improvement.

Reviewer #4: (No Response)

**Editorial and Data Presentation Modifications?**

Reviewer #2: Based on the authors’ responses to peer reviews and revised content, key modifications to editorial standards and data presentation are summarized.

Reviewer #4: (No Response)

**Summary and General Comments**

Reviewer #2: This study advances DSS prediction by transforming vague clinical criteria into quantifiable metrics. The scoring system enables precision triage, optimizing resource allocation and improving patient outcomes. Future work should focus on external validation, inclusion of pediatric/adolescent populations, and assessment of seasonal or geographic variability in DSS kinetics.

The manuscript demonstrates strong scientific merit, with rigorous methods and clear clinical applicability, though broader validation is warranted before widespread adoption.

Reviewer #4: Thank you for addressing my questions and concerns. I suggest that the authors recommend applying the DSS score as early as day 2 or day 3, as you mentioned in your response to the previous question, and that this recommendation be included at the end of the discussion section or in the conclusion.

PLOS authors have the option to publish the peer review history of their article (what does this mean?). If published, this will include your full peer review and any attached files.). If published, this will include your full peer review and any attached files.). If published, this will include your full peer review and any attached files.). If published, this will include your full peer review and any attached files.

...

Reviewer #2: No

Reviewer #4: No

 **Figure resubmission:**While revising your submission, we strongly recommend that you use PLOS’s NAAS tool (https://ngplosjournals.pagemajik.ai/artanalysis) to test your figure files. NAAS can convert your figure files to the TIFF file type and meet basic requirements (such as print size, resolution), or provide you with a report on issues that do not meet our requirements and that NAAS cannot fix.While revising your submission, we strongly recommend that you use PLOS’s NAAS tool (https://ngplosjournals.pagemajik.ai/artanalysis) to test your figure files. NAAS can convert your figure files to the TIFF file type and meet basic requirements (such as print size, resolution), or provide you with a report on issues that do not meet our requirements and that NAAS cannot fix.

After uploading your figures to PLOS’s NAAS tool - https://ngplosjournals.pagemajik.ai/artanalysis, NAAS will process the files provided and display the results in the "Uploaded Files" section of the page as the processing is complete. If the uploaded figures meet our requirements (or NAAS is able to fix the files to meet our requirements), the figure will be marked as "fixed" above. If NAAS is unable to fix the files, a red "failed" label will appear above. When NAAS has confirmed that the figure files meet our requirements, please download the file via the download option, and include these NAAS processed figure files when submitting your revised manuscript.**Reproducibility:**To enhance the reproducibility of your results, we recommend that authors of applicable studies deposit laboratory protocols in protocols.io, where a protocol can be assigned its own identifier (DOI) such that it can be cited independently in the future. Additionally, PLOS ONE offers an option to publish peer-reviewed clinical study protocols. Read more information on sharing protocols at https://plos.org/protocols?utm_medium=editorial-email&utm_source=authorletters&utm_campaign=protocolsTo enhance the reproducibility of your results, we recommend that authors of applicable studies deposit laboratory protocols in protocols.io, where a protocol can be assigned its own identifier (DOI) such that it can be cited independently in the future. Additionally, PLOS ONE offers an option to publish peer-reviewed clinical study protocols. Read more information on sharing protocols at https://plos.org/protocols?utm_medium=editorial-email&utm_source=authorletters&utm_campaign=protocols

---

## [Decision Letter · Decision Letter 2]

10 Apr 2026

Dear Dr Ho Dang Trung,

We are pleased to inform you that your manuscript 'Quantifying the Kinetics of Hematocrit and Platelet Count During Febrile Phase to Develop a Scoring System for Predicting Dengue Shock Syndrome in Adults: A Matched Case-Control Study from a Hospital in Viet Nam' has been provisionally accepted for publication in PLOS Neglected Tropical Diseases.

Best regards,

Mohammad Jokar, DVM

Guest Editor

Sujatha Sunil

Section Editor

Shaden Kamhawi

co-Editor-in-Chief

Paul Brindley

co-Editor-in-Chief

Reviewer's Responses to Questions

**Key Review Criteria Required for Acceptance?**

**Methods**

-Are the objectives of the study clearly articulated with a clear testable hypothesis stated?

-Is the study design appropriate to address the stated objectives?

-Is the population clearly described and appropriate for the hypothesis being tested?

-Is the sample size sufficient to ensure adequate power to address the hypothesis being tested?

-Were correct statistical analysis used to support conclusions?

-Are there concerns about ethical or regulatory requirements being met?

Reviewer #4: (No Response)

**Results**

-Does the analysis presented match the analysis plan?

-Are the results clearly and completely presented?

-Are the figures (Tables, Images) of sufficient quality for clarity?

Reviewer #4: (No Response)

**Conclusions**

-Are the conclusions supported by the data presented?

-Are the limitations of analysis clearly described?

-Do the authors discuss how these data can be helpful to advance our understanding of the topic under study?

-Is public health relevance addressed?

Reviewer #4: (No Response)

**Editorial and Data Presentation Modifications?**

Reviewer #4: (No Response)

**Summary and General Comments**

Reviewer #4: (No Response)

PLOS authors have the option to publish the peer review history of their article (what does this mean?). If published, this will include your full peer review and any attached files.). If published, this will include your full peer review and any attached files.). If published, this will include your full peer review and any attached files.). If published, this will include your full peer review and any attached files.

...

Reviewer #4: No

---

## [Editor Report · Acceptance letter]

Dear Dr Ho Dang Trung,

We are delighted to inform you that your manuscript, "Quantifying the Kinetics of Hematocrit and Platelet Count During Febrile Phase to Develop a Scoring System for Predicting Dengue Shock Syndrome in Adults: A Matched Case-Control Study from a Hospital in Viet Nam," has been formally accepted for publication in PLOS Neglected Tropical Diseases.

Best regards,

Shaden Kamhawi

co-Editor-in-Chief

Paul Brindley

co-Editor-in-Chief
